# Associations between Phosphate Concentrations and Hospital Mortality in Critically Ill Patients Receiving Mechanical Ventilation

**DOI:** 10.3390/jcm11071897

**Published:** 2022-03-29

**Authors:** Beong Ki Kim, Chi Young Kim, Sua Kim, Yu Jin Kim, Seung Heon Lee, Je Hyeong Kim

**Affiliations:** 1Division of Pulmonary, Allergy, and Critical Care Medicine, Department of Internal Medicine, Korea University Ansan Hospital, Ansan 15355, Korea; negero55@korea.ac.kr (B.K.K.); himazinn@naver.com (C.Y.K.); gene2001@naver.com (Y.J.K.); kucmnfs@hanmail.net (S.H.L.); 2Department of Critical Care Medicine, Korea University Ansan Hospital, Ansan 15355, Korea; sua0047@gmail.com

**Keywords:** hyperphosphatemia, hypophosphatemia, phosphate, hospital mortality, respiration, artificial, critical care

## Abstract

Phosphate concentrations change continuously throughout hospitalization; however, it is unclear which available phosphate measures are most clinically important for predicting hospital mortality. Therefore, we investigated phosphate concentrations in association with hospital mortality following admission to the intensive care unit. We retrospectively enrolled all adult patients receiving mechanical ventilation. Phosphate concentrations were divided into three categories: initially measured phosphate (iP); maximum–minimum phosphate values (ΔP); and phosphate arithmetic average (Pmean). In total, 175 patients were enrolled. The hospital mortality rate was 32.6%, and the most common primary diagnosis was respiratory failure. In multivariable logistic regression analyses, the odds ratios for hospital mortality in association with ΔP and Pmean values were 1.56 and 2.13, respectively (*p* < 0.0001). According to the obtained receiver operating characteristic curve, ΔP (0.75) and Pmean (0.72) each showed a fair predictive power for hospital mortality. In evaluating relative risks, we found that higher concentrations of Pmean and ΔP were each associated with a higher hospital mortality. ΔP and Pmean values were significantly associated with hospital mortality in critically ill patients, compared to iP. These findings showed that throughout hospitalization, it is important to reduce phosphate level fluctuations and maintain appropriate phosphate concentrations through consistent monitoring and corrections.

## 1. Introduction

Phosphate is an important intracellular anion and is a major component of the lipid bilayer, where it acts as a physiologic buffer. Phosphate also plays vital roles in a range of physiological processes, including energy storage via adenosine triphosphate and oxygen transport via 2,3–diphosphoglycerate (found in red blood cells) [1]. Therefore, it is critical to maintain proper phosphate concentrations. This is accomplished through renal clearance, intestinal absorption, and storage in the bone [2]. In hospitalized patients with acute illnesses, phosphate homeostasis is not well controlled, thus frequently resulting in phosphate imbalance. The reported incidence of hypo- and hyperphosphatemia in all hospitalized patients is approximately 9.1% and 11.6%, respectively [3]. Phosphate imbalance is known to occur more commonly in critically ill patients, with hypo- and hyperphosphatemia reported at rates of 20% and 45%, respectively [4].

Clinical manifestations of phosphate imbalance likewise vary widely, ranging from mild to life-threatening presentations. Phosphate imbalance can lead to tissue hypoxia, decreased myocardial contractility, arrhythmia, metabolic encephalopathy, and death [5,6,7,8]. Hypophosphatemia causes respiratory muscle weakness, decreased diaphragmatic contractility, and failure to wean from mechanical ventilation [9,10,11]. For these reasons, phosphate imbalance is associated with prolonged respiratory failure and an increase in the length of stay (LOS) in the intensive care unit (ICU) [12,13]. The effect of phosphate imbalance has a greater impact on prognoses for critically ill patients, especially those undergoing mechanical ventilation [14,15,16,17,18].

Previous studies have reported that phosphate imbalance at admission is associated with hospital mortality in critically ill patients [19,20,21]. However, changes in phosphate concentrations occur continuously throughout hospitalization, because the tendency to maintain homeostasis is not controllable; these shifts vary according to the patient’s condition. Patients with initial hypophosphatemia may develop hyperphosphatemia during hospitalization, and vice versa. Importantly, phosphate imbalance affects the function of respiratory muscles such as the diaphragm [9]. Thus, we hypothesized that even small changes in phosphate levels among critically ill patients undergoing mechanical ventilation can lead to profound effects on prognoses. These small changes can be induced by various phosphate parameters, including minimal, maximal, and average concentrations. Currently, there is a study showing that the greater phosphate change, the higher the in-hospital mortality rate [22]. However, there are no studies on patients undergoing mechanical ventilation, and little is known about which aforementioned phosphate parameters may be highly associated with hospital mortality. We hypothesized that the initially measured phosphate concentration has a greater effect on the patient’s in-hospital mortality than the change or average value of phosphate. Therefore, we conducted an exploratory retrospective and observational study to evaluate associations between various phosphate measures and hospital mortality in critically ill patients undergoing mechanical ventilation.

## 2. Materials and Methods

### 2.1. Study Design and Population

We retrospectively reviewed the medical records of patients who were admitted to the medical ICU at a single university-affiliated hospital in Korea from 1 December 2018 to 30 November 2019. All enrolled patients were adults over 18 years of age, had been admitted to the ICU for medical illness, and received mechanical ventilation during their stay. Pregnant women, those with a medical history of parathyroid disease, dialysis patients, patients receiving mechanical ventilation for less than 24 h, and those admitted with do-not-resuscitate orders (including do-not-intubate orders) were excluded from the current study. We also excluded patients who did not have their phosphate concentrations measured within 24 h of their ICU admission or at least twice during their ICU stay. We calculated each phosphate measure as one continuous hospitalization for patients who were admitted to the ICU more than twice during the same hospitalization for the same morbidity. We calculated phosphate levels separately in cases of rehospitalization after discharge.

### 2.2. Data Collection and Variable Definitions

Based on a retrospective review of medical records, we collected data on basic demographic characteristics, reasons for hospital admission, Acute Physiology and Chronic Health Evaluation II (APACHE II) scores, and Charlson comorbidity index for each patient. Phosphate concentrations were divided into three categories: (1) initial phosphate (iP), the first phosphate concentration measured within 24 h of ICU admission; (2) delta phosphate (ΔP), the range of change in phosphate levels (i.e., the concentration obtained by subtracting the minimum phosphate value from the maximum phosphate value during the ICU stay); and (3) mean phosphate levels (Pmean), the arithmetical mean concentrations of all measured phosphate concentrations throughout the ICU stay. All phosphate concentrations were measured via a direct ultraviolet spectrophotometry method using a Cobas 8000 modular analyzer (Roshe, Milan, Italy). The reference range used for normal phosphate concentrations was 2.5–4.8 mg/dL (1.03–1.98 mmol/L). The concentrations of ionized calcium (reference range 4.5–5.3 mg/dL), potassium (reference range 3.5–5.5 mmol/L), magnesium (reference range 1.6–2.6 mg/dL), and albumin (reference range 3.5–5.2 g/dL) were measured, as were the body mass index (BMI, kg/m^2^), PaO_2_/FiO_2_ ratio (the ratio of arterial oxygen partial pressure to fractional inspired oxygen, mmHg), creatinine levels (reference range 0.5–1.2 mg/dL), and creatinine clearance (mL/min/1.73 m^2^) calculated via the Cockcroft–Gault equation. The primary outcome of our study was hospital mortality, and we also recorded the duration of mechanical ventilation; the mean LOS in the ICU and in the hospital; and 28-day and ICU-specific mortality rates. The study protocol was approved by the hospital institutional review board (IRB No. 2020AS0008), which waived the requirement for informed consent due to the retrospective nature of this study. This research was carried out in accordance with the principles of the Declaration of Helsinki and its later amendments.

### 2.3. Statistical Analyses

Categorical variables were expressed as numbers and percentages, and continuous variables were expressed as means and standard deviations as appropriate. We conducted univariable and multivariable logistic regression analyses to determine effects on hospital mortality according to the three phosphate categories. A receiver operating characteristic (ROC) curve was evaluated to determine the area under the curve (AUC) as well as the sensitivity and specificity for each measure. We used the Youden index to find the optimal cut-off values for predicting hospital mortality. Positive predictive values (PPV) and negative predictive values (NPV) were also calculated to obtain additional information on mortality predictions. Log odds ratios were obtained to evaluate the relative risk of mortality for each phosphate concentration category. All statistical analyses were performed using SAS statistical software (ver. 9.4; SAS institute, Cary, NC, USA) and R statistical software (ver. 4.0.3; The R Foundation, Vienna, Austria). A two-tailed *p*-value less than 0.05 was considered statistically significant.

## 3. Results

### 3.1. Baseline Characteristics and Clinical Course

In total, 175 consecutively presenting patients were enrolled during the study period (Figure 1). Among the enrolled patients, 61.7% (*n* = 108) were men, and the mean age was 66.68 ± 16.36 years. The most common primary cause of hospitalization was respiratory failure (*n* = 49, 28.0%). The mean BMI of the enrolled patients was 22.05 ± 4.50 kg/m^2^, and the mean values for the APACHE II score, the Charlson comorbidity index, and the PaO_2_/FiO_2_ ratio were 28.35 ± 7.06, 4.46 ± 2.48, and 223.57 ± 145.82 mmHg, respectively. The mean creatinine clearance rate was 67.68 ± 43.98 mL/min/1.73 m^2^, and the proportion of patients presenting with chronic kidney disease without dialysis was 5.7% (*n* = 10) (Table 1).

The mean duration of mechanical ventilation was 11.97 ± 11.37 days, the mean LOS in the ICU was 15.82 ± 12.60 days, and the total mean in-hospital LOS was 29.83 ± 20.01 days. The 28-day, ICU, and hospital mortality rates were 28.0% (*n* = 49), 29.1% (*n* = 51), and 32.6% (*n* = 57), respectively (Table 2).

### 3.2. Phosphate Concentrations and Mortality

As shown in Table 1, a total of 107 (61.1%) enrolled patients were supplemented with intravenous phosphate at least once during the course of their hospitalization; the mean number of phosphate applications was 2.86 ± 3.55. According to the results of the univariable logistic regression analyses, we found a statistically significant association between Pmean values and hospital mortality (odds ratio (OR), 2.24; 95% confidence interval (CI) 1.63–3.06, *p* < 0.0001). The OR for Pmean was 2.13 (95% CI 1.52–2.99, *p* < 0.0001) in multivariable analyses after adjusting for age, APACHE II scores, BMI, creatinine clearance, and PaO_2_/FiO_2_ ratio. The AUC for Pmean in the evaluated ROC curve was 0.72 (95% CI 0.63–0.82), and hospital mortality was predicted with a sensitivity of 57.9%, a specificity of 89.0%, a PPV of 71.7%, and an NPV of 81.4% when Pmean exceeded 3.70 mg/dL.

Next, we found that the mean value for ΔP was 2.50 ± 1.58 mg/dL. In univariable and multivariable analyses, the ORs for ΔP were 1.65 (95% CI 1.35–2.01, *p* < 0.0001) and 1.56 (95% CI 1.28–1.91, *p* < 0.0001), respectively. The AUC of ΔP was 0.75 (95% CI 0.67–0.82), and hospital mortality was predicted with a sensitivity of 24.6%, a specificity of 97.5%, a PPV of 82.4%, and an NPV of 72.8% when ΔP exceeded 5.58 mg/dL. Additionally, we found that the mean iP value was 3.23 ± 1.67 mg/dL. The OR for iP was statistically significant in the univariable analysis (1.31, 95% CI 1.07–1.59, *p* = 0.01), but not in the multivariable analysis (Table 3 and Table 4, Figure 2).

We additionally provided a hospital mortality 2 × 2 table calculated based on the cut-off value obtained through the Youden index (Appendix A).

Finally, we constructed the log odds graphs showing the relative risk of hospital mortality for each phosphate category. The log odds graph for iP showed a U-shaped pattern; mortality was lowest between 2 and 3 mg/dL. In contrast, we found that the slope for Pmean gradually increased when phosphate concentrations were above 3 mg/dL, and that the slope for ΔP increased continuously (Figure 3).

## 4. Discussion

We conducted this study to determine the phosphate values (iP, ΔP, Pmean) most strongly associated with hospital mortality in critically ill patients undergoing mechanical ventilation. Our results showed that ΔP and Pmean were associated with hospital mortality and were more strongly associated with hospital mortality, compared with iP. According to the ROC curve, ΔP and Pmean values each showed good predictive power for mortality. Moreover, in the log odds graphs, we found that mortality risk increased as the mean phosphate concentrations exceeded the normal range and as the fluctuation range of phosphate concentrations increased.

In previous study, the relative risk for mortality according to phosphate concentrations was found to be approximately one to two times higher for abnormal phosphate concentrations, compared with referent values [21]. These findings were similar to our results, and showed that ΔP and Pmean are associated with hospital mortality. Furthermore, the AUC for ΔP and Pmean ranged from 0.7 to 0.8, thus showing good predictive power. The sensitivity and specificity values calculated via the Youden index were also good. Hence, we concluded that phosphate concentrations showed good potential as a tool for predicting mortality. Additional studies are required to find the most appropriate measurement intervals and frequencies and to determine predictive phosphate concentrations.

Our results suggested that ΔP and Pmean affect hospital mortality to a greater degree, compared with iP. Therefore, according to these results and the findings of prior studies, we suggest that careful correction of phosphate concentrations via frequent monitoring throughout the ICU stay can help improve the patient’s prognosis. In addition, our finding that hospital mortality increased with increasing ΔP and Pmean levels suggested that it is necessary to reduce phosphate fluctuations and maintain stability during hospitalization.

When evaluating Pmean values, a U-shaped graph was expected based on previous results (similar to the result seen for iP in the current investigation) showing increased mortality rates in the presence of hypophosphatemia and hyperphosphatemia [19,22]. However, contrary to our expectations, up to 2 mg/dL, it seemed to remain constantly in a slightly decreasing trend, and then increased linearly at concentrations above 2 mg/dL. Previous findings, similar to our result, showed linear associations between phosphate concentrations and mortality [23,24]. Thus, additional research is required to confirm these results in future intensive investigations.

Phosphate levels have been shown to affect patient prognoses within a range of studies [25,26,27,28]. Similar to prior research, our study also revealed that phosphate levels are an important factor that influenced hospital mortality. However, it seems that physicians tend to overlook the importance of phosphate concentrations when evaluating mortality risk. According to a nationwide survey conducted in the Netherlands, less than half of patients (46%) had their phosphate concentrations measured on a daily basis during their hospitalization, and only 39% of patients had their phosphate concentrations measured at the time of hospital admission [29]. In the current study, 42.7% of patients did not have their phosphate levels measured within 24 h of admission. Moreover, 38.9% of the enrolled patients were not provided any level of intravenous phosphate supplementation, and patients receiving supplementation received a mean of only 2.86 doses during their ICU stay (mean LOS, 15.82 days). Previous studies have shown that the dosage and speed of phosphate supplementation varies widely, and that phosphate levels can be corrected safely and effectively [29,30,31]. To our knowledge, few studies have evaluated the effects of direct phosphate supplementation and/or dosage corrections on patient prognoses. However, our results indicate that minimizing phosphate fluctuations and constantly correcting mean phosphate concentrations to the normal range will be helpful in improving patient prognoses.

The strengths of our study were as follows. First, our findings demonstrated the importance of maintaining a proper phosphate level and reducing phosphate fluctuations, rather than only measuring initial phosphate levels. This is an important finding for all physicians in charge of critically ill patients. It is important to constantly calibrate phosphate levels through daily monitoring. However, it is necessary to validate this finding in highly-powered investigations. Nonetheless, our findings emphasized that many physicians overlook this important point, and more attention should therefore be given to phosphate monitoring in order to improve the prognoses of critically ill patients in the future.

In addition to the substantial strengths of our investigation, we acknowledge the following study limitations. First, we conducted a retrospective single-center study. Consequently, missing values inherent in this type of study were likely to have affected the obtained results and generalizability of our findings. Second, we obtained mean phosphate concentrations using the arithmetic mean (vs. the geometric mean). Thus, phosphate dosing intervals were not considered, which may have affected the obtained results. Third, measuring the patients’ phosphate levels only twice during the course of their hospitalization could not accurately reflect the true means and ranges of phosphate concentrations in a typical hospital setting, which may have affected our results. Finally, the effects of the method of dietary supplementation (e.g., enteral or parenteral nutrition) were not investigated in this or prior studies; hence, the associated impacts on phosphate concentrations are unknown. These questions and limitations should be evaluated more thoroughly in future intensive investigations.

## 5. Conclusions

Our study demonstrated that ΔP and Pmean levels were associated with hospital mortality and were more strongly correlated with mortality, compared with iP. Moreover, ΔP and Pmean each showed good predictive power with respect to mortality. We concluded that it is important to reduce fluctuations in phosphate levels during hospitalization and to maintain mean phosphate concentrations within the normal range in order to help improve patient prognoses. However, because physicians tend to overlook the importance of these metrics, more attention should be given to phosphate concentrations through daily monitoring and correction, and the importance of this factor in medical training and continuing medical education should be emphasized. Thus, our findings inform research directions and, if confirmed, will ultimately inform medical guidelines and effective clinical decision making.

## Figures and Tables

**Figure 1 jcm-11-01897-f001:**
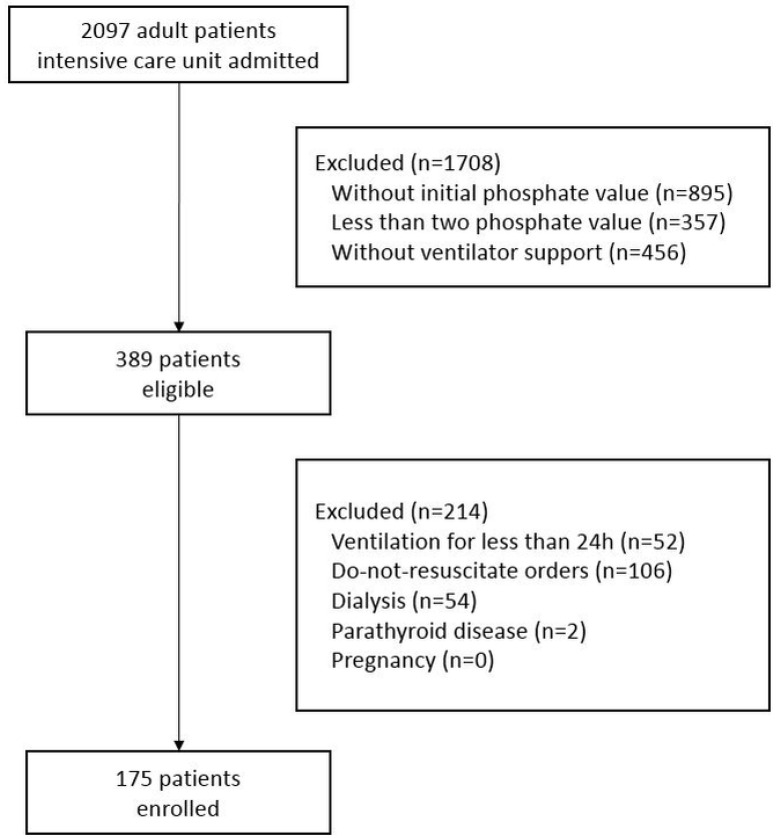
The flow chart of patient registration.

**Figure 2 jcm-11-01897-f002:**
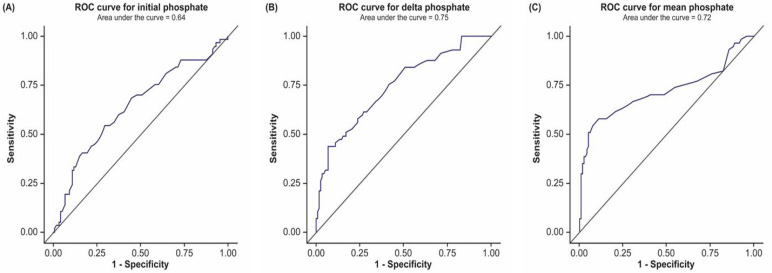
Receiver operating characteristic (ROC) curve according to phosphate metric categories. (**A**) Initial phosphate: the first phosphate value measured within 24 h of admission to the intensive care unit (ICU). (**B**) Delta phosphate: the range of change obtained by subtracting the minimum phosphate concentration from the maximum phosphate concentration measured during the ICU stay. (**C**) Mean phosphate: the arithmetical mean value for phosphate concentrations measured during the ICU stay.

**Figure 3 jcm-11-01897-f003:**
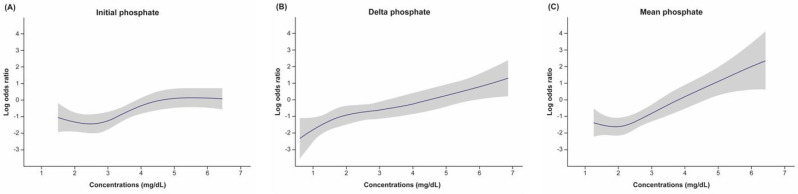
Log odds graphs evaluating hospital mortality risk according to phosphate values in each category. The shading indicates 95% confidence intervals. (**A**) Initial phosphate: the first phosphate value measured within 24 h of admission to the intensive care unit (ICU). (**B**) Delta phosphate: the range of change obtained by subtracting the minimum phosphate concentration from the maximum phosphate concentration measured during the ICU stay. (**C**) Mean phosphate: the arithmetic mean value for phosphate concentrations measured during the ICU stay.

**Table 1 jcm-11-01897-t001:** Baseline patient medical and demographic characteristics (*n* = 175).

Patient Characteristic	Value
Sex (male)	108 (61.7%)
Age (years)	66.68 ± 16.36
Main diagnosis	
Respiratory failure	49 (28.0%)
Septic shock	46 (26.3%)
Aspiration	25 (14.3%)
Post-cardiac arrest syndrome	21 (12.0%)
Hypovolemic shock	9 (5.1%)
Neurologic disease	9 (5.1%)
Cardiovascular disease	5 (2.9%)
Others	11 (6.3%)
Body mass index (kg/m^2^)	22.05 ± 4.50
Phosphate categories (mg/dL)	
Initial phosphate levels *	3.23 ± 1.67
Delta phosphate levels **	2.50 ± 1.58
Mean phosphate levels ^†^	2.75 ± 0.57
Phosphate intravenous supplementation	
Total number of supplies	2.86 ± 3.55
Proportion of phosphate supply	107 (61.1%)
Creatinine (mg/dL)	1.03 ± 0.54
Creatinine clearance (mL/min/1.73 m^2^)	67.68 ± 43.98
Chronic kidney disease without dialysis	10 (5.7%)
Potassium (mmol/L)	3.83 ± 0.70
Ionized calcium (mg/dL)	4.40 ± 0.45
Magnesium (mg/dL)	2.04 ± 0.37
Albumin (g/dL)	3.03 ± 0.54
APACHE II score	28.35 ± 7.06
Charlson comorbidity index	4.46 ± 2.48
PaO_2_/FiO_2_ ratio (mmHg)	223.57 ± 145.82

Note: APACHE II, Acute Physiology and Chronic Health Evaluation II; ICU, intensive care unit; LOS, length of stay; PaO_2_/FiO_2_ ratio, the ratio of arterial oxygen partial pressure to fractional inspired oxygen. Data are presented as counts (%) or as means ± standard deviations as appropriate. * Initial phosphate: the first phosphate value within 24 h of admission to the ICU. ** Delta phosphate: the range of change obtained by subtracting the minimum phosphate concentration from the maximum phosphate concentration measured during the ICU stay. ^†^ Mean phosphate: the arithmetical mean value for phosphate concentrations measured during the ICU stay.

**Table 2 jcm-11-01897-t002:** Patient outcomes such as length of stay and mortality (*n* = 175).

Patient Outcome	Value
Duration of mechanical ventilation (days)	11.97 ± 11.37
LOS in the ICU (days)	15.82 ± 12.60
LOS in the hospital (days)	29.83 ± 20.01
28-day mortality	49 (28.0%)
ICU mortality	51 (29.1%)
Hospital mortality	57 (32.6%)

Note: LOS, length of stay; ICU, intensive care unit.

**Table 3 jcm-11-01897-t003:** Logistic regression analyses for phosphate categories in association with hospital mortality.

	Univariable			Multivariable *		
Phosphate	Odds Ratio	95% CI	*p*-Value	Odds Ratio	95% CI	*p*-Value
Initial phosphate **	1.31	1.07–1.59	0.01	1.15	0.92–1.42	0.06
Delta phosphate ^†^	1.65	1.35–2.01	<0.0001	1.56	1.28–1.91	<0.0001
Mean phosphate ^††^	2.24	1.63–3.06	<0.0001	2.13	1.52–2.99	<0.0001

Note: APACHE II, Acute Physiology and Chronic Health Evaluation II; BMI, body mass index; CI, confidence interval; PaO_2_/FiO_2_ ratio, the ratio of arterial oxygen partial pressure to fractional inspired oxygen; ICU, intensive care unit. * Multivariable models were adjusted for age, APACHE II scores, BMI, creatinine clearance, and the PaO_2_/FiO_2_ ratio. ** Initial phosphate: the first phosphate value measured within 24 h of admission to the ICU. ^†^ Delta phosphate: the range of change obtained by subtracting the minimum phosphate concentration from the maximum phosphate concentration measured during the ICU stay. ^††^ Mean phosphate: the arithmetical mean value for phosphate concentrations measured during the ICU stay.

**Table 4 jcm-11-01897-t004:** Hospital mortality predictions according to phosphate metric categories calculated by a receiver operating characteristic curve.

Phosphate	AUC	95% CI	Criterion (mg/dL)	Sensitivity	Specificity	PPV	NPV
Initial phosphate *	0.64	0.55–0.73	>3.80	49.1%	72.9%	46.7%	74.8%
Delta phosphate **	0.75	0.67–0.82	>5.58	24.6%	97.5%	82.4%	72.8%
Mean phosphate ^†^	0.72	0.63–0.82	>3.70	57.9%	89.0%	71.7%	81.4%

Note: AUC, area under the curve; CI, confidence interval; NPV, negative predictive value; PPV, positive predictive value; ICU, intensive care unit. * Initial phosphate: the first phosphate value measured within 24 h of admission to the ICU. ** Delta phosphate: the range of change obtained by subtracting the minimum phosphate concentration from the maximum phosphate concentration measured during the ICU stay. ^†^ Mean phosphate: the arithmetical mean value for phosphate concentrations measured during the ICU stay.

## Data Availability

The data used for this study, though not available in a public repository, will be made available to other researchers upon reasonable request.

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
