# Peer review of "Associations between Phosphate Concentrations and Hospital Mortality in Critically Ill Patients Receiving Mechanical Ventilation"

_jcm, 2022, doi:10.3390/jcm11071897_

Round 1

Reviewer 1 Report

Thank you for the opportunity to review this study.
The authors present a single center retrospective study with 175 patients hospitalised in the ICU and under mechanical ventilation and investigate the association between three categories of phosphate measurements and mortality. The deltaP and mortality has already been studied in larger studies in non-ICU hospitalised patients (28,149 patients in the study of Thongprayoon et al, 2020). Although the topic is of major interest, I am concerned that this low-powered, single-center retrospective study with some methodological weaknesses does not add solid knowledge to the current literature on the topic. All of my following recommendations are intended to strengthen the study.

Abstract :

-Lines 15-16: Pmean could be unclear for the reader discovering the abstract without having read the full article. I would suggest the Authors to clarify this abbreviation.

-Lines 24-25: As discussed later in this Review, I am not sure of this conclusion. The Authors highlight an association between deltaP and mortality. However, phosphate fluctuation could only be a marker of severity and not directly associated to mortality. In this hypothesis, it might seem too strong to conclude that physicians should actively avoid phosphates fluctuations.

Introduction:

-Lines 45-47 : Please add references for this sentence. Furthermore, the association between phosphate imbalance and mortality is not so clear. Some studies didn’t find such an association. (Federspiel et al. for example DOI: 10.1111/aas.13136 ).

-Lines 52-53 : Please specify why phosphate levels might change during ICU stay.

-Lines 54-58 : This part lacks references and might not be as clear as that in the literature.

-Lines 61: As the model used is a logistic regression, I would suggest the Authors to speak of an association rather than a correlation.

-Lines 60-62: I would suggest the Author to better clarify the exact research question and the outcomes that will be measured. I would also suggest the Author to discuss in the introduction the study of Thongprayoon et al, 2020 (DOI 10.1186/s12882-020-02090-3) which studied phosphates fluctuations in an hospital setting and its association with mortality.  

Methods :

-Lines 72-74: I will suggest the Authors to add a study flow-chart to clarify patient’s enrollment.

-Line 74 : Please clarify, what the Authors meant with « integrative phosphate measures ».

-Lines 81 : Charlson comorbidity “indices” should be changed to “index”.

-Line 84: The Authors have chosen to use deltaP, which is defined by the Authors as the concentration obtained by subtracting the minimum phosphate value from the maximum phosphate value during the ICU stay. The study of Thongprayoon et al chose to analyse the trend of phosphate changes (i.e. if phosphate levels were increasing or decreasing during time). It could have also been interesting to add this information in this current study.

-Lines 89-90 : The Author chose a reference range for normal phosphate levels of 2.5-4.8mg/dL. As several studies chose different cut-offs (Federspiel et al ( DOI: 10.1111/aas.13136, Broman et al DOI: 10.1213/ANE.0000000000002077, Wang et al https://doi.org/10.1186/s12871-019-0746-2 )., the comparison to results of these previous studies might be a compromised.  Can the author explain the choice of this cut-off value?

-Lines 90-92 : Please add the units for all the electrolytes and laboratory values.

-94-95 : Since the Authors have chosen to present a ROC curve with a best cut-off value to predict hospital mortality by using the Youden Index, I think that a 2x2 table with sensitivity and specificity could be very informative in the results section or in the supplementary tables.

-Line 113 : Please complete the statistical analysis section by precising  if the p value chosen was bilateral (two-tailed p value).

Results :

I found the presentation of the results a little unclear. I would suggest the author to separate the first table into two separate tables: 1) Patients characteristics and 2) the primary outcome: mortality at several time point. The text describing the results should follow the tables. 

-Table 1 : Please, add the unit for P/F ratio (also in the main text).

Creatinine, potassium, magnesium etc are found under the subsection “phosphate supplementation” which makes the table confusing.

Hospital mortality in the table 1 is 32.6%. However, in the Abstract the Author mention 38,9%. This point should be verified.

It is not described in the method section that phosphate supplementation was going to be looked for and neither how phosphate was supplemented (oral or parenteral route, dose). So, the number and proportion of supplies is a little unclear. 

The title of the table is somewhat imprecise as the table does not only present the baseline characteristics but also the outcomes.

-Lines 139-140: please add where this information can be found in the tables (table 1).

-line 144 : The Authors chose not to categorize phosphates levels into subgroups. Were each phosphate levels (admission, delta and mean) log-linear to be able to perform the logistic regression that way? It would have been interesting to have the mean higher and lower phosphates levels during ICU stay.

-Lines 144-146 : Why did the Authors chose to adjust to age, APACHE II, BMI, creatinine clearance and P/F ratios ? Please, also specify in the methodology. Were these variables categorized to do the analysis? If not, were they log-linear?
I would also suggest the authors no to adjust to age as age is already included in the APACHE II score which could lead to collinearity.

-Line 185: log odds graphs were not specified in the methods sections. This should be completed in the methodology section. The Authors should also specifiy why they had to do the log odds instead of keeping the odds only.

Discussion

General comment: I feel that due to the study design and the small sample size, the Author should be careful with the way they discuss their results. I would suggest to avoid expressions such as “risk factors” and I would suggest to favor “association” instead. I would also suggest to avoid strong statement such as “very” or “extremely”.

-Line 199: as this is a retrospective observational study assessing the association between phosphates levels and mortality through a logistic regression analysis, I would suggest the author to speak of associated factor rather than risk factor to mortality. 

-Line 205 : the Authors compare their studies to several studies and conclude that these studies found the same results and that Pmean and deltaP are associated with mortality. However, some of the cited studies have investigated the association between phosphate levels on ICU admission and mortality (Wang et al, Cheungpasitporn et al) and are thus not comparable.

-Lines 212-213 : This sentence is somehow unclear. Could the Authors please clarify.

-Lines 219-220 : I am not sure that this study allows to conclude that phosphate fluctuations should be avoided. Several other interpretations could be proposed: 1) maybe, it is just extreme values who influence the Pmean and deltaP who are associated to mortality and not the fluctuations themselves, 2) Another explanation could be that hypoP and hyperP are both associated to mortality and that if you combine both factors the mortality risk increases, 3) it is also possible that phosphate fluctuations are just a marker of severity but not directly associated to mortality

-Lines 217-219 : The author conclude that, as mortality seems to be associated to increased deltaP, it is necessary to reduce phosphate fluctuations during hospitalization. As phosphates supplementation and changes in phosphate levels after supplementation have not been assessed in this study, I would suggest not to draw such a direct conclusion. Further studies are warranted on phosphate substitution and possible impact on mortality to be able to draw this kind of conclusion.  

-Lines 227-233 :  Authors give new information on rates of hypophosphatemia in the discussion. I would suggest not to give new results in the discussion section that have not been described in the results section.

-Lines 227-238: References should be added.  

-Lines 230-232: How and when phosphate substitution has been administered is not detailed in the study results. I would suggest to reevaluate this assumption after adding this information in the results sections.

-Lines: 242-243 : The Author speak of a current study with 2,097 patients enrolled and screened. It is not known what the source of this number is. Please clarifiy.

-Lines 250-252: Please add references.

-Lines 271-272: Limitation section: I would say that more importantly, phosphate supplementation has not been fully investigated which would have been very interesting and useful to better explain the study results. Information on episodes of hypophosphatemia or hyperphosphatemia are also not described and these last have been associated to poor outcomes in previous studies too (Suzuki et al). The mean and delta phosphate level could also be questionable as we do not have the information of when the phosphate has been measured in the course of the ICU stay (early or late). A high interindividual variability in terms of number of phosphate measurement and supplementation is a possibility too and would alter the interpretation of the results. It would have been interesting to have the number of in-hospital serum phosphate measurements.

Author Response

March 14, 2022

Dear Editor and Reviewer(s) of Journal of Clinical Medicine

We are resubmitting our manuscript, the manuscript ID jcm-1615706 entitled " Associations between Phosphate Concentrations and Hospital Mortality in Critically Ill Patients receiving Mechanical Ventilation".

We appreciate your acknowledgement of the value of our manuscript’s subject and your offer to reconsider our manuscript with revision. We also agree with the comments raised by the reviewers, and have made revisions accordingly.

We have provided a point-by-point response to the issues that were raised by the reviewers and have also added the revised portions to our manuscript highlighted by track changes.

We believe that the revision will resolve the concerns raised by the reviewers and hope that our manuscript will now be accepted for publication. If you have any further questions or suggestions, please do not hesitate to contact us.

Thank you for considering our manuscript for publication in Journal of Clinical Medicine.

We look forward to hearing from you.

Sincerely yours,

Je Hyeong Kim, MD, PhD.

Responses to Reviewer's Comments to Author

We appreciate your excellent review of our manuscript. Your valuable comments helped us to make a better revision.

=============================================================================

REVIEWER 1

Comments to the Author:

Abstract :

-Lines 15-16: Pmean could be unclear for the reader discovering the abstract without having read the full article. I would suggest the Authors to clarify this abbreviation.

[Answer] Thank you for your comments. We added some clarification; “Pmean, (mean phosphate- arithmetical average)”.

-Lines 24-25: As discussed later in this Review, I am not sure of this conclusion. The Authors highlight an association between deltaP and mortality. However, phosphate fluctuation could only be a marker of severity and not directly associated to mortality. In this hypothesis, it might seem too strong to conclude that physicians should actively avoid phosphates fluctuations.

[Answer] Thanks for the valuable comments. We agree with the reviewer’s opinion. In our study, reducing phosphate fluctuation and maintaining an appropriate phosphate concentration during hospitalization decreased the in-hospital mortality. As the reviewer said, this may not be a direct influencing factor to the mortality. Although factors that could affect this results were adjusted, it was mentioned in the conclusion section that additional high powered studies were needed due to the nature of a small retrospective observational study.

From the study limitations (lines 292-306):

We acknowledge the following study limitations. First, we conducted a retrospective single-center study. Consequently, missing values inherent to this type of study are likely to have affected the obtained results and generalizability of our findings. Second, we obtained mean phosphate concentrations using the arithmetic mean (vs. the geometric mean). Thus, phosphate dosing intervals were not considered, which may have affected the obtained results. Third, measuring the patients’ phosphate levels only twice during the course of their hospitalization cannot accurately reflect the true means and ranges of phosphate concentrations in a typical hospital setting, which may have affected our results. Finally, the effects of the method of dietary supplementation, e.g., enteral or parenteral nutrition) have not been investigated in this or prior studies; hence, the associated impacts on phosphate concentrations are unknown. These questions and limitations should be evaluated more thoroughly in future high-powered investigations.

Introduction:

-Lines 45-47: Please add references for this sentence. Furthermore, the association between phosphate imbalance and mortality is not so clear. Some studies didn’t find such an association. (Federspiel et al. for example DOI: 10.1111/aas.13136 ).

[Answer] Thanks for the good comments. We added the reference as you pointed out (References 12,13). The association between phosphate imbalance and mortality was as controversial as the reviewer said, so the phrase was deleted.

-Lines 52-53: Please specify why phosphate levels might change during ICU stay.

[Answer] Line 53: We have added the following sentence to make the content clearer; “because the tendency to maintain homeostasis is out of control.”

-Lines 54-55: This part lacks references and might not be as clear as that in the literature.

[Answer] Line 56. We added the study of TR Gravelyn et al. as a reference (Reference [9]).

-Lines 61: As the model used is a logistic regression, I would suggest the Authors to speak of an association rather than a correlation.

[Answer] Thank you for your careful consideration of terminology. We have corrected the terminology as you mentioned.

-Lines 60-62: I would suggest the Author to better clarify the exact research question and the outcomes that will be measured. I would also suggest the Author discuss in the introduction the study of Thongprayoon et al, 2020 (DOI 10.1186/s12882-020-02090-3) which studied phosphates fluctuations in an hospital setting and its association with mortality.

[Answer] Lines 56-60. We introduced the study you mentioned as you advised so that our research could be expressed more clearly, and we made modifications as follows; “Thus, we hypothesized that even small changes in phosphate levels among critically ill patients undergoing mechanical ventilation can lead to profound effects on the prognoses. These small changes can be induced by various phosphate parameters, including minimal, maximal, and average concentrations.

Lines 60-68. Currently, there is a study that shows that the greater phosphate change, the higher the in-hospital mortality rate [22]. However, there are no studies on patients undergoing mechanical ventilation, and little is known about which aforementioned phosphate parameters may be highly associated with hospital mortality. Therefore, we conducted an exploratory study to evaluate associations between various phosphate measures and hospital mortality in critically ill patients undergoing mechanical ventilation.”

Methods :

-Lines 72-74: I will suggest the Authors to add a study flow-chart to clarify patient’s enrollment.

[Answer] We have additionally provided a flow-chart of patient enrollment as Figure 1 in the Results section.

-Line 74: Please clarify, what the Authors meant with «integrative phosphate measures».

[Answer] The meaning of the phrase you pointed out was expressed more clearly as follows; “We calculated each phosphate measure as one continuous hospitalization for patients who were admitted to the ICU more than twice during the same hospitalization for the same morbidity.”(Lines 80-82)

-Line 81.Charlson comorbidity “indices” should be changed to “index”.

[Answer] As you mentioned, we modified it with an “index” .(Line 87)

-Line 84: The Authors have chosen to use deltaP, which is defined by the Authors as the concentration obtained by subtracting the minimum phosphate value from the maximum phosphate value during the ICU stay. The study of Thongprayoon et al chose to analyse the trend of phosphate changes (i.e. if phosphate levels were increasing or decreasing during time). It could have also been interesting to add this information in this current study.

[Answer] Thank you for introducing the interesting topic. We conducted the analysis using the methods used in the study you mentioned. However, our study could not obtain any statistically significant results because a relatively small number of patients were enrolled compared to the study of Thongprayoon et al. The results of our analysis with the help of biostatistician at our institution are shown as follows.

Phosphate change

Numbers (n=175)

In-hospital mortality

OR (95% CI)

P value

≤ -2.8

21

5 (23.8%)

As reference

-2.7 to -2.1

13

3 (23.1%)

0.96 (0.19 – 4.92)

0.96

-2.0 to -1.4

20

5 (25.0%)

1.07 (0.26 – 4.44)

0.93

-1.3 to -0.7

17

3 (17.7%)

0.69 (0.14 - 3.40)

0.64

-0.6 to 0

7

0 (0.0%)

N/A

0.99

0.1 to 0.6

6

0 (0.0%)

N/A

0.99

0.7 to 1.3

16

4 (25.0%)

1.07 (0.24 – 4.84)

0.93

1.4 to 2.0

19

3 (15.8%)

0.60 (0.12 – 2.94)

0.53

2.1 to 2.7

16

6 (37.5%)

1.92 (0.46 – 7.99)

0.37

≥ 2.8

40

28 (70.0%)

7.47 (2.23 – 25.06)

0.001

-Lines 89-90: The Author chose a reference range for normal phosphate levels of 2.5-4.8mg/dL. As several studies chose different cut-offs (Federspiel et al ( DOI: 10.1111/aas.13136, Broman et al DOI: 10.1213/ANE.0000000000002077, Wang et al https://doi.org/10.1186/s12871-019-0746-2 ), the comparison to results of these previous studies might be a compromised. Can the author explain the choice of this cut-off value?

[Answer] Thank you for reviewing my manuscript in detail. As you mentioned, the studies have used various cut-off values for the normal range. The difference in values is thought to be due to the phosphate measuring instruments and methods. The cut-off value was introduced according to the measurement methods used in our institution and the protocol of the instrument. And, although the studies you mentioned compare the results by dividing each category for hypo-, normo- and hyperphosphatemia, however, our study is different from the study mentioned in the analysis method, because it did not compare the results by classifying phosphate categories. Therefore, I think there is a difference between the studies you mentioned and our study.

-Lines 90-92: Please add the units for all the electrolytes and laboratory values.

[Answer] Thank you for the good advice. We added the normal range and unit for all the electrolytes and laboratory values (Lines 96-103).

-94-95: Since the Authors have chosen to present a ROC curve with a best cut-off value to predict hospital mortality by using the Youden Index, I think that a 2x2 table with sensitivity and specificity could be very informative in the results section or in the supplementary tables.

[Answer] Thank you for the valuable comments. We tried to create a 2x2 table based on the best cut-off value as your comments were given.

Death (n=118)

Survival (n=57)

Total (n=175)

Initial phosphate value

Survival

86 (72.9%)

29 (50.9%)

115 (65.7%)

Death

32 (27.1%)

28 (49.1%)

60 (34.3%)

Delta phosphate value

Survival

115 (97.5%)

43 (75.4%)

158 (90.3%)

Death

3 (2.5%)

14 (24.6%)

17 (9.7%)

Mean phosphate value

Survival

104 (88.1%)

24 (42.1%)

128 (73.1%)

Death

14 (11.9%)

33 (57.9%)

47 (26.9%)

(Supplementary Table 1) The 2x2 table of hospital mortality predicted by cut-off values calculated by Youden index

-Line 113: Please complete the statistical analysis section by precising if the p value chosen was bilateral (two-tailed p value).

[Answer] We provided additional information in the statistical section as follows; “A two-tailed p-value less than 0.05 was considered statistically significant.” (Line 122)

Results

I found the presentation of the results a little unclear. I would suggest the author to separate the first table into two separate tables: 1) Patients characteristics and 2) the primary outcome: mortality at several time point. The text describing the results should follow the tables. 

[Answer] Thank you for the comments. As you mentioned, the table was separated, and the text describing the results placed after table.

-Table 1: Please, add the unit for P/F ratio (also in the main text).

[Answer] We added the unit for P/F ratio.

Creatinine, potassium, magnesium .etc are found under the subsection “phosphate supplementation” which makes the table confusing.

[Answer] The section has been re-arranged to avoid confusion.

Hospital mortality in the table 1 is 32.6%. However, in the Abstract the Author mention 38,9%. This point should be verified.

[Answer] Thank you for finding the important error. The mortality in the abstract was corrected.

It is not described in the method section that phosphate supplementation was going to be looked for and neither how phosphate was supplemented (oral or parenteral route, dose). So, the number and proportion of supplies is a little unclear. 

[Answer] Thank you for your valuable comments. In our institution, phosphate supplementation is provided only intravenously, we added that in the main text and table. However, in our study, what we wanted to know was not the relationship between the exact dose, route, rate of administration and the results, but how many physicians overlooked the importance of phosphate. Therefore, we did not collect such data.

The title of the table is somewhat imprecise as the table does not only present the baseline characteristics but also the outcomes.

[Answer] As you mentioned above, the table was presented as a table divided into baseline characteristics and outcomes.

-Lines 139-140: please add where this information can be found in the tables (Table 1).

[Answer] We added the phases where the information can be found.

-Line 144: The Authors chose not to categorize phosphates levels into subgroups. Were each phosphate levels (admission, delta and mean) log-linear to be able to perform the logistic regression that way? It would have been interesting to have the mean higher and lower phosphates levels during ICU stay.

[Answer] We performed all statistical analyses used in our study in consultation with a biostatistician at our institution. As a results, based on the opinion that log-linear is satisfied, we assumed log-linear and performed the analysis. And thank you for your suggestions on additional interesting topics. Although we could not include it in our study this time, it gave us a lot of inspiration for the next research idea, and if there is a chance, we would like to do a future analysis on this.

-Lines 144-146 Why did the Authors choose to adjust for age, APACHE II, BMI, creatinine clearance and P/F ratios ? Please, also specify in the methodology. Were these variables categorized to do the analysis? If not, were they log-linear?
I would also suggest the authors not adjust for age as age is already included in the APACHE II score which could lead to collinearity.

[Answer] Thank you for your professional advice. We tried to select the adjustment variables that were previously known to be related to mortality. As mentioned earlier, after consulting with a biostatistician, we checked and calculated the variance-inflation and generalized variance-inflation factors (VIFs and GVIFs) for linear, generalized linear, and other regression models to confirm that there is no multicollinearity and proceeded with the analysis.

-Line 185: log odds graphs were not specified in the methods sections. This should be completed in the methodology section. The Authors should also specify why they had to do the log odds instead of keeping the odds only.

 [Answer] We mentioned the log odds graph in the statistical analysis section as follows; “Log odds ratios were obtained to evaluate the relative risk of mortality for each phosphate concentration category.” (Lines 118-119). We wanted to intuitively check the relationship between each categories of phosphate and hospital mortality. For this reason, we thought that the log odds ratio was more appropriate than the odds ratio graphs only.

Discussion

General comment: I feel that due to the study design and the small sample size, the Author should be careful with the way they discuss their results. I would suggest to avoid expressions such as “risk factors” and I would suggest to favor “association” instead. I would also suggest to avoid strong statement such as “very” or “extremely.”

[Answer] Thank you for the good comments. We avoided too strong expressions like “very” and “extremely”. And the “risk factor” was modified to the expression of “association”.

-Line 199: as this is a retrospective observational study assessing the association between phosphates levels and mortality through a logistic regression analysis, I would suggest the author to speak of associated factor rather than risk factor to mortality. 

[Answer] As mentioned earlier, “risk factor” was changed to “association.”

-Line 205. The Authors compare their studies to several studies and conclude that these studies found the same results and that Pmean and deltaP are associated with mortality. However, some of the cited studies have investigated the association between phosphate levels on ICU admission and mortality (Wang et al, Cheungpasitporn et al) and are thus not comparable.

[Answer] Thank you for your detailed review. The references you mentioned have been removed.

-Lines 212-213. This sentence is somehow unclear. Could the Authors please clarify.

[Answer] We deleted the sentence because the meaning may not be clear.

-Lines 219-220: I am not sure that this study allows to conclude that phosphate fluctuations should be avoided. Several other interpretations could be proposed: 1) maybe, it is just extreme values who influence the Pmean and deltaP who are associated to mortality and not the fluctuations themselves, 2) Another explanation could be that hypoP and hyperP are both associated to mortality and that if you combine both factors the mortality risk increases, 3) it is also possible that phosphate fluctuations are just a marker of severity but not directly associated to mortality.

[Anser] Thank you for your approach through various interpretations of our results. As you said, the fluctuations of phosphate may not be a factor that directly affects the in-hospital mortality of patients. However, as shown in the aforementioned study by Thongprayoon et al., fluctuation may be related to in-hospital mortality, we presented the results with the expression “suggest.”

-Lines 217-219: The authors conclude that, as mortality seems to be associated to increased deltaP, it is necessary to reduce phosphate fluctuations during hospitalization. As phosphates supplementation and changes in phosphate levels after supplementation have not been assessed in this study, I would suggest not to draw such a direct conclusion. Further studies are warranted on phosphate substitution and possible impact on mortality to be able to draw this kind of conclusion

[Answer] Thanks for the advice. We modified the sentence as follows; “Therefore, according to these results and the findings of prior studies, we suggested that careful correction of phosphate concentrations via frequent monitoring throughout the ICU stay can help improve the patient’s prognosis.” (Lines 253-255)

-Lines 227-233Authors give new information on rates of hypophosphatemia in the discussion. I would suggest not to give new results in the discussion section that have not been described in the results section.

[Answer] New information that dis not provided in the results section has been deleted.

-Lines 227-238: References should be added.

[Answer] We have deleted some of the content you mentioned.

-Lines 230-232: How and when phosphate substitution has been administered is not detailed in the study results. I would suggest to reevaluate this assumption after adding this information in the results sections.

[Answer] Also, the part you mentioned has been deleted.

-Lines: 242-243 : The Authors speak of a current study with 2,097 patients enrolled and screened. It is not known what the source of this number is. Please clarify.

[Answer] As pointed out earlier, we added a flow-chart as Figure 1 to the Results section (Lines above 140)

-Lines 250-252: Please add references.

[Answer] The meaning of the previous studies was not clear, so that part was deleted.

-Lines 271-272: Limitation section: I would say that more importantly, phosphate supplementation has not been fully investigated which would have been very interesting and useful to better explain the study results. Information on episodes of hypophosphatemia or hyperphosphatemia are also not described and these last have been associated to poor outcomes in previous studies too (Suzuki et al). The mean and delta phosphate level could also be questionable as we do not have the information of when the phosphate has been measured in the course of the ICU stay (early or late). A high interindividual variability in terms of number of phosphate measurement and supplementation is a possibility too and would alter the interpretation of the results. It would have been interesting to have the number of in-hospital serum phosphate measurements.

[Answer] Thank you for your valuable comments of our manuscript. Professional and detailed review has helped more of our manuscript to revise for the better. The advice you mentioned gave us a lot of inspiration for the future research direction.

Reviewer 2 Report

Thank you for invinting me to review this manuscript. The authors assessed the phospate level in ICU patients who need invasivbe mechanical ventilation. The manuscript is easy to ready and well described. however I have some comments.

- I don' t understand why you used the Pmean with patients who received phosphate durign ICU stay (61% of the patients had phosphate supply during ICU stay). I can't undertand what could represent this value in clinical setting

-phosphate levels depends on several variables as dialysis, nutrition. All those variables were not described durig the ICU.

-For such alow number of patients (n=175), result may be expressed with median value and IQR.

Author Response

March 14, 2022

Dear Editor and Reviewer(s) of Journal of Clinical Medicine

We are resubmitting our manuscript, the manuscript ID jcm-1615706 entitled " Associations between Phosphate Concentrations and Hospital Mortality in Critically Ill Patients receiving Mechanical Ventilation".

We appreciate your acknowledgement of the value of our manuscript’s subject and your offer to reconsider our manuscript with revision. We also agree with the comments raised by the reviewers, and have made revisions accordingly.

We have provided a point-by-point response to the issues that were raised by the reviewers and have also added the revised portions to our manuscript highlighted by track changes.

We believe that the revision will resolve the concerns raised by the reviewers and hope that our manuscript will now be accepted for publication. If you have any further questions or suggestions, please do not hesitate to contact us.

Thank you for considering our manuscript for publication in Journal of Clinical Medicine.

We look forward to hearing from you.

Sincerely yours,

Je Hyeong Kim, MD, PhD.

Responses to Reviewer's Comments to Author

We appreciate your excellent review of our manuscript. Your valuable comments helped us to make a better revision.

=============================================================================

REVIEWER 2

Comments to the Author:

Thank you for inviting me to review this manuscript. The authors assessed the phospate level in ICU patients who need invasive mechanical ventilation. The manuscript is easy to ready and well described. however I have some comments.

I don' t understand why you used the Pmean with patients who received phosphate during ICU stay (61% of the patients had phosphate supply during ICU stay). I can't understand what could represent this value in clinical setting.

[Answer] Thank you for your valuable comments. We want to show that phosphate levels are an important factor influencing a patient’s prognosis, but many doctors overlook its importance. Therefore, although the normal phospate level in our institution is 2.5–4.8 mg/dL and the mean phosphate levels (Pmean) for patients enrolled our study during the ICU stay is at the lower limit of 2.75 ± 0.57 mg/dL. However, about 39% of patients did not receive phosphate, and the patients receiving the phosphate supplementation is only 2.86 doses. We have presented the results for the purpose of highlighting this point.

-Phosphate levels depends on several variables as dialysis, nutrition. All those variables were not described during the ICU.

[Answer] Thank you for the comments on the important point. We were also aware of the importance of nutrition. Therefore, this point was mentioned in the limitation. And although dialysis is a very important factor that affects the phosphate level, but we excluded patients on dialysis because we considered this to be an exogenous variable that was difficult to control.

For such a low number of patients (n=175), results may be expressed with median value and IQR.

[Answer] Thank you for the detailed comments. We discussed the statistical method used in our study with a biostatistician at our institution. After thorough normality tests, we decided to use the mean and standard deviation rather than the median and IQR.

Reviewer 3 Report

Thank you for the opportunity to be involved in the revision of this manuscript. The authors performed a retrospective observational study investigating the association between different phosphate concentration measurements, i.e., initial phosphate, delta phosphate, and mean phosphate levels, and hospital mortality in critically ill mechanically ventilated patients. The strengths of this study are that it is overall well written and that it addresses an important, yet probably still overlooked, clinical question. I would suggest the following minor revisions.

Abstract

  • Line 16: “Pmean” should be defined.

Introduction

  • Line 45: “weaning failure”. The authors should specify they mean weaning from mechanical ventilation.
  • Lines 60-62: I would suggest the author specify what their a priori hypothesis is, e.g., that average phosphate levels are more associated with mortality, as compared to the other phosphate concentration measures. Moreover, the authors should specify which variables are the independent variables. It is clear their dependent variable is hospital mortality, however, the independent variables are only briefly mentioned and not clearly specified as initial phosphate, mean phosphate, and delta phosphate.
  • Line 60: The authors should specify that the study was retrospective and observational.

Materials and methods

  • Any protocol for phosphate replacement or correction?
  • Line 68: “applied”. What do the authors mean by “applied”? Please consider deleting this word, if not necessary.
  • Lines 94-96: I would suggest the authors specify that hospital mortality is the primary outcome, whereas the other outcome variables are secondary outcomes, if that is the case.
  • Line 104: How were the variables selected for inclusion in the multivariate analysis? Was the selection based on p-value? Was it stepwise, forward, backward, etc.? The authors should specify this. How did the authors deal with multicollinearity? If they only included age, APACHE II scores, BMI, creatinine clearance, and PaO2/FiO2 ratio as potentially confounding variables (as one would expect from the Results section) and no other baseline variables, they should specify this in the Methods. Finally, they should clarify whether or not they included in the multivariate analysis only those baseline variables that were significantly associated with the outcome in the univariate analysis or they included them directly in the multivariate analysis.

Results

  • Line 117: How many patients were screened? Which were the reasons for exclusion?
  • Paragraph 3.1: Please round all the numerical values appropriately, e.g., age, APACHE II, Charlson, days.
  • Table 1: The authors should divide this table in two tables, one with baseline and demographic variables and the other with phosphate levels, phosphate supplementation, and outcome variables. I would recommend including all baseline variables together, e.g., creatinine, creatinine clearance, CKD without dialysis, potassium, calcium, magnesium, albumin, APACHE II, Charlson, P/F ratio seem baseline variable, whereas many other variables in Table 1 are phosphate measures collected during the hospital course and outcome variables.
  • Line 141: Why is Pmean value reported within the brackets referring to the mean number of phosphate applications? Please report the values on its own.
  • Duration of mechanical ventilation and LOS are reported among the outcomes, although they are not primary outcome. Would the authors consider providing further analysis on the relationship between the phosphate measurement and these secondary outcomes as supplementary material? This could be interesting, given the fact that they often mentioned the possible impact of phosphate levels on respiratory muscle dysfunction.

Discussion

  • Lines 202-204: If the reference range used for normal phosphate concentrations is 2.5–4.8 mg/dL (lines 89-90), and “we found that the slope for Pmean gradually increased when phosphate concentrations were above 3 mg/dL” (lines 187-188), I do not understand why “mortality risk increased as the mean phosphate concentrations exceeded the normal range”.
  • Lines 205-206: “was found to be approximately one to two times higher”. Something like “was found to be approximately one to two times higher for abnormal phosphate concentrations, compared with…” would make this sentence more clear.
  • Line 211: The use of the word “excellent” contrasts with the more cautious “good” in the previous lines, which I think is more appropriate, considering the study results.
  • Lines 212-213: If I understood the study results correctly, the specificity only tell us, among patients who survive, how many had ΔP values lower than 5.58 mg/dL This is not very useful, because, considering only specificity, we would not know how many patients, among those with ΔP values exceeding 5.58 mg/dL, will die. From this perspective, PPV is more useful and is quite high (82.4%). However, the authors should comment on the fact that Delta phosphate may not be very informative, as one of the two values may be collected near the time of death, hence too late to help improve patient outcome.
  • Lines 216-221: “our finding that hospital mortality increased with increasing ΔP levels suggests that it is necessary to reduce phosphate fluctuations during hospitalization.” The fact that the relationship between mean phosphate concentration and log odds ratio for mortality is not linear (increased mortality below about 1.8 and above about 2.2 mg/dL) may support the conclusion that phosphate levels should be maintained stable. What are the authors’ thoughts on that?
  • Lines 225-226: The graph shape seems to be U-shaped. Moreover, it seems to me that the slope is constant in a smaller range than the one the authors report, i.e., 1.8-2.2 mg/dl, more or less.
  • Line 230: Do the authors have data on this or is this just a hypothesis?
  • Lines 231-232: Are the authors implying that phosphate levels should not be too aggressively corrected?
  • Lines 242-247: These data should be reported in the Results sections and only referred to here.

Author Response

March 14, 2022

Dear Editor and Reviewer(s) of Journal of Clinical Medicine

We are resubmitting our manuscript, the manuscript ID jcm-1615706 entitled " Associations between Phosphate Concentrations and Hospital Mortality in Critically Ill Patients receiving Mechanical Ventilation".

We appreciate your acknowledgement of the value of our manuscript’s subject and your offer to reconsider our manuscript with revision. We also agree with the comments raised by the reviewers, and have made revisions accordingly.

We have provided a point-by-point response to the issues that were raised by the reviewers and have also added the revised portions to our manuscript highlighted by track changes.

We believe that the revision will resolve the concerns raised by the reviewers and hope that our manuscript will now be accepted for publication. If you have any further questions or suggestions, please do not hesitate to contact us.

Thank you for considering our manuscript for publication in Journal of Clinical Medicine.

We look forward to hearing from you.

Sincerely yours,

Je Hyeong Kim, MD, PhD.

Responses to Reviewer's Comments to Author

We appreciate your excellent review of our manuscript. Your valuable comments helped us to make a better revision.

=============================================================================

REVIEWER 3

Comments to the Author:

Thank you for the opportunity to be involved in the revision of this manuscript. The authors performed a retrospective observational study investigating the association between different phosphate concentration measurements, i.e., initial phosphate, delta phosphate, and mean phosphate levels, and hospital mortality in critically ill mechanically ventilated patients. The strengths of this study are that it is overall well written and that it addresses an important, yet probably still overlooked, clinical question. I would suggest the following minor revisions.

Abstract

Line 16: “Pmean” should be defined.

 [Answer] Thank you for the good comments. We have defined Pmean more clearly.(Line 16).

Introduction

Line 45: “weaning failure”. The authors should specify they mean weaning from mechanical ventilation.

[Answer] Thank you for the advice. We clarified the meaning of “weaning failure” as follows; “failure to wean from mechanical ventilation”. (Line 45)

Lines 60-62: I would suggest the author specify what their a priori hypothesis is, e.g., that average phosphate levels are more associated with mortality, as compared to the other phosphate concentration measures. Moreover, the authors should specify which variables are the independent variables. It is clear their dependent variable is hospital mortality, however, the independent variables are only briefly mentioned and not clearly specified as initial phosphate, mean phosphate, and delta phosphate.

[Answer] Thank you for the advice on the detailed point. We added more to clarify about the hypothesis and independent variables of our study as follows; “ We hypothesized that the initially measured phosphate concentration have a great effect on the patient’s in-hospital mortality than the change or average value of phosphate. Therefore, we conducted an exploratory retrospective and observational study to evaluate associations between various phosphate measures and hospital mortality in critically ill patients undergoing mechanical ventilation.” (Lines 64-68)

Line 60: The authors should specify that the study was retrospective and observational.

 [Answer] We added that our study was a retrospective and observational design.(Line 66)

Materials and methods

Any protocol for phosphate replacement or correction?

[Answer] Because our study was a retrospective observational study, there was no specific protocol for this. Depending on the preference and decision of the attending physician, the supplementation and route of phosphate was determined.

Line 68: “applied”. What do the authors mean by “applied”? Please consider deleting this word, if not necessary.

[Answer] As you said, “applied” has been removed.

Lines 94-96: I would suggest the authors specify that hospital mortality is the primary outcome, whereas the other outcome variables are secondary outcomes, if that is the case.

[Answer] Thank you for your valuable advice. We specify the primary outcome more clearly as follows; “The primary outcome of our study was hospital mortality, and we also recorded the duration of mechanical ventilation; the mean LOS in the ICU and in the hospital; and 28-day and ICU-specific mortality rates.”(Lines 102-105)

Line 104: How were the variables selected for inclusion in the multivariate analysis? Was the selection based on p-value? Was it stepwise, forward, backward, etc.? The authors should specify this. How did the authors deal with multicollinearity? If they only included age, APACHE II scores, BMI, creatinine clearance, and PaO2/FiO2 ratio as potentially confounding variables (as one would expect from the Results section) and no other baseline variables, they should specify this in the Methods. Finally, they should clarify whether or not they included in the multivariate analysis only those baseline variables that were significantly associated with the outcome in the univariate analysis or they included them directly in the multivariate analysis.

[Answer] Thank you for your professional comments. We performed statistical analysis of this study with the advice of biostatistician in our institution. He advised us to designate around five adjustment variables in consideration of the percentage of dependent variables. And we included variables known to be highly correlated with mortality in previous studies in full- model analysis rather than stepwise, forward and backward. Additionally, we checked Calculates variance-inflation and generalized variance-inflation factors (VIFs and GVIFs) for linear, generalized linear, and other regression models for confirmation that there is no multicollinearity.

Results

Line 117: How many patients were screened? Which were the reasons for exclusion?

[Answer] A total of 2,097 patients were admitted to the ICU during the study period, and 175 patients were finally analyzed, excluding those who met the exclusion criteria. Patients with parathyroid disease and undergoing dialysis were excluded because we thought that they were exogenous variables that were difficult to control. Pregnant women were an exclusion criteria at the time of initial study planning, but there were no patients on the enrolled subject, and patients with do-not-resuscitate orders were excluded as they were thought to have a significant influence on the dependent variable. More details are in Figure 1, added at the request of another reviewer.

Paragraph 3.1: Please round all the numerical values appropriately, e.g., age, APACHE II, Charlson, days.

[Answer] There is no mention of the decimal point in the author’s guideline, so it is expressed up to the second decimal point. If you have specific requests, please let us know.

Table 1: The authors should divide this table in two tables, one with baseline and demographic variables and the other with phosphate levels, phosphate supplementation, and outcome variables. I would recommend including all baseline variables together, e.g., creatinine, creatinine clearance, CKD without dialysis, potassium, calcium, magnesium, albumin, APACHE II, Charlson, P/F ratio seem baseline variable, whereas many other variables in Table 1 are phosphate measures collected during the hospital course and outcome variables.

[Answer] The other reviewer also gave a similar opinion, so we divided the table into the baseline characteristics’ part and outcome part.

Line 141: Why is Pmean value reported within the brackets referring to the mean number of phosphate applications? Please report the values on its own.

[Answer] We reported the value itself.

The Duration of mechanical ventilation and LOS are reported among the outcomes, although they are not the primary outcome. Would the authors consider providing further analysis on the relationship between the phosphate measurement and these secondary outcomes as supplementary material? This could be interesting, given the fact that they often mentioned the possible impact of phosphate levels on respiratory muscle dysfunction.

[Answer] Thank you for the advice on an interesting topic.Firstly, the mean and standard deviation of the mechanical ventilation, LOS of ICU, and LOS of hospital were compared between the survivors and the non-survivors. Next, we categorized each phosphate based on the cut-off value obtained by the Youden index and compared the same variables. Unfortunately, the results were not significantly different.

MV day

P

ICU day

P

Hospital day

P

Survivor (n=118)

10.44±10.43

0.43

14.21±12.76

0.69

27.86±22.28

0.06

Non-survivor (n=57)

12.02±13.08

13.35±13.77

21.60±19.39

Cut-off value

MV day

P

ICU day

P

Hospital day

P

iP

> 3.80 (n=60)

12.87±13.08

0.14

15.35±14.58

0.33

25.88±23.91

0.98

≤ 3.80 (n=115)

9.96±10.25

13.19±12.20

25.78±20.28

deltaP

> 5.58 (n=17)

15.00±12.84

0.18

18.18±14.95

0.23

29.94±27.11

0.51

≤ 5.58 (n=158

10.52±11.13

13.47±12.82

25.37±20.89

Pmean

> 3.70 (n=46)

11.76±12.95

0.61

13.13±14.00

0.65

23.72±22.33

0.45

≤ 3.70 (n=129)

10.67±10.76

14.22±12.76

26.57±21.27

Discussion

Lines 202-204: If the reference range used for normal phosphate concentrations is 2.5–4.8 mg/dL (lines 89-90), and “we found that the slope for Pmean gradually increased when phosphate concentrations were above 3 mg/dL” (lines 187-188), I do not understand why “mortality risk increased as the mean phosphate concentrations exceeded the normal range”.

[Answer] Because the methods of our study was a retrospective observational study, we cannot know the cause in detail. However, it is known that hyperphosphatemia can cause a cardiovascular event (Block GA et al., 2004 and Slinin Y et al., 2005) and there are also results showing that hyperphosphatemia is an independent risk factor for mortality (Dominik G. Haider et al., 2015). So we think our study did show similar results.

Lines 205-206: “was found to be approximately one to two times higher”. Something like “was found to be approximately one to two times higher for abnormal phosphate concentrations, compared with…” would make this sentence clearer. (Lines 241-243)

[Answer] Thank you for the advice to clarify the meaning of the paragraph.

Line 211: The use of the word “excellent” contrasts with the more cautious “good” in the previous lines, which I think is more appropriate, considering the study results.

[Answer] Thank you for the advice on the use of words in reference to the results of the study. We corrected it as advised.

Lines 212-213: If I understood the study results correctly, the specificity only tell us, among patients who survive, how many had ΔP values lower than 5.58 mg/dL This is not very useful, because, considering only specificity, we would not know how many patients, among those with ΔP values exceeding 5.58 mg/dL, will die. From this perspective, PPV is more useful and is quite high (82.4%). However, the authors should comment on the fact that Delta phosphate may not be very informative, as one of the two values may be collected near the time of death, hence too late to help improve patient outcome.

[Answer] Thank you for the professional comments. Similar comments were raised by other reviewers, so we decided to remove the sentences.

Lines 216-221: “our finding that hospital mortality increased with increasing ΔP levels suggests that it is necessary to reduce phosphate fluctuations during hospitalization.” The fact that the relationship between mean phosphate concentration and log odds ratio for mortality is not linear (increased mortality below about 1.8 and above about 2.2 mg/dL) may support the conclusion that phosphate levels should be maintained stable. What are the authors’ thoughts on that?

[Answer] Thank you for your valuable comments. We agree with the reviewer’s comments and revised as follows; “In addition, our finding that hospital mortality increased with increasing ΔP and Pmean levels suggests that it is necessary to reduce phosphate fluctuations and maintain stability during hospitalization.”(Lines 255-257).

Lines 225-226: The graph shape seems to be U-shaped. Moreover, it seems to me that the slope is constant in a smaller range than the one the authors report, i.e., 1.8-2.2 mg/dl, more or less.

[Answer] We actually expected a definite U-shaped graph as in the study of W.Cheungpasitporn et al. (https://doi.org/10.1080/21548331.2018.1483172). However the slope was kept relatively constant, so we described it as such in the manuscript. We revised the text following the reviewer’s opinion as follows; “However, contrary to our expectations, up to 2 mg/dL, it seems to remain constantly in a slightly decreasing trend, and then increased linearly at concentrations above 2 mg/dL”.(Lines 261-262)

Line 230: Do the authors have data on this or is this just a hypothesis?

[Answer] As this study is a retrospective observational study, it was not possible to supply phosphate according to the exact protocol. In our institution, phosphate is supplied for decision by the physician in charge. Therefore, we hypothesized that if the phosphate level was low, phosphate would be supplied intravenously more frequently. However, another reviewer also pointed out this part, so we decided to delete the text.

Lines 231-232: Are the authors implying that phosphate levels should not be too aggressively corrected?

[Answer] It is not like that. An excessively high phosphate level has resulted in poor prognosis in previous studies (Dominik G. Haider et al., 2015 and B Kestenbaum et al., 2005) and ours as well, but we think it is necessary to reduce fluctuations and maintain an appropriate level through continuous monitoring during hospitalization, As we said before, this part has been deleted.

Lines 242-247: These data should be reported in the Results sections and only referred to here.

[Answer] Following the reviewer’s advice, we moved the data to Figure 1 and revised the sentence as follows; “In the current study, 42.7% of patients did not have their phosphate levels measured within 24 h of admission”. (Lines 274-275)

Round 2

Reviewer 2 Report

thank you , I have no more question

Author Response

We appreciate your detailed review of our manuscript and the opportunity to revise it for the better.